# Decoding EEG in Motor Imagery Tasks with Graph Semi-Supervised Broad Learning

**Qingshan She ***, **Yukai Zhou, Haitao Gan, Yuliang Ma and Zhizeng Luo**

School of Automation, Hangzhou Dianzi University, Hangzhou, Zhejiang 310018, China;
ykzhou@hdu.edu.cn (Y.Z.); htgan@hdu.edu.cn (H.G.); mayuliang@hdu.edu.cn (Y.M.); luo@hdu.edu.cn (Z.L.)
* Correspondence: qsshe@hdu.edu.cn; Tel.: +86-571-8691-9130

**Abstract:** In recent years, the accurate and real-time classification of electroencephalogram (EEG) signals has drawn increasing attention in the application of brain-computer interface technology (BCI). Supervised methods used to classify EEG signals have gotten satisfactory results. However, unlabeled samples are more frequent than labeled samples, so how to simultaneously utilize limited labeled samples and many unlabeled samples becomes a research hotspot. In this paper, we propose a new graph-based semi-supervised broad learning system (GSS-BLS), which combines the graph label propagation method to obtain pseudo-labels and then trains the GSS-BLS classifier together with other labeled samples. Three BCI competition datasets are used to assess the GSS-BLS approach and five comparison algorithms: BLS, ELM, HELM, LapSVM and SMIR. The experimental results show that GSS-BLS achieves satisfying Cohen's kappa values in three datasets. GSS-BLS achieves the better results of each subject in the 2-class and 4-class datasets and has significant improvements compared with original BLS except subject C6. Therefore, the proposed GSS-BLS is an effective semi-supervised algorithm for classifying EEG signals.

**Keywords:** brain-computer interface; electroencephalogram; semi-supervised learning; broad learning system; graph label propagation

## 1. Introduction

The Brain-Computer Interface (BCI) is a technology that only needs to use the signals generated by the human brain when subjected to specific stimuli to control external devices or systems [1], which is independent of normal peripheral neuromuscular channels. In recent years, the application of BCI technology has become more and more extensive, which has achieved fruitful results in the fields of games, rehabilitation, and aerospace [2]. BCI is mainly used to accurately detect the patient's intention of exercise in the field of active motor rehabilitation, so the patients can actively participate in the process of exercise training and induce neural plasticity [3]. This is due to the low cost of electroencephalogram (EEG) signals acquisition, ease of use, and minimal side effects in the subjects. The measured EEG signals are translated into a command for an application by three general steps: the first step is pre-processing EEG signals, second is extracting features from these signals, and the last is classifying EEG features. However, EEG signals often have characteristics of low signal-to-noise ratio, time-varying, and instability [4]. As a result, it remains a challenging task to achieve accurate and real-time classification of EEG signals.

There are many kinds of machine learning algorithms to effectively identify different types of EEG signals. The support vector machine (SVM) [5,6] maps data samples to high-dimensional space through kernel functions and learns to obtain a hyperplane to classify the samples. K-Nearest Neighbor (KNN) [7] discriminates samples by calculating distances, for instance, Euclidean distances. The extreme learning machine (ELM) [8] is a single hidden layer neural network. The input layer and

hidden layer connection weights are randomly generated and do not need to be adjusted. The hidden layer and output layer connection weights can be obtained by the least-squares method, so it is efficient and in real-time. In recent years, deep learning (DL) has also been applied to the classification of EEG signals. Li et al. [9] combine with multi-fractal attributes to construct a deep learning model based on denoising encoders to identify different motion imaging tasks. In [10], the spatiotemporal characteristics of EEG signals are considered. They use stacked automatic encoders and convolutional neural networks to classify EEG signals, and further propose a new input form by extracting time, frequency, and position information. Their approach yields a 9% improvement over the winning algorithm of the competition. An et al. [11] use deep belief net (DBN) to train a weak classifier and borrow the idea of the Ada-boost algorithm to combine the trained weak classifiers as a more powerful one. This is an improvement of 4%–6% compared with SVM.

However, deep learning requires complex structural adjustments and complicated calculations during training. Aiming at such problems, Professor Chen proposes a broad learning system (BLS) approach [12]. The essence of BLS is a random vector function links neural network (RVFLNN). First, the raw data is mapped to mapping features (MF) by random weights. Next, the feature nodes are mapped to enhanced nodes (EN) as a width extension in a similar way. It aims to enhance the nonlinearity of the model to obtain better results. Finally, the feature nodes and the enhancement nodes are simultaneously mapped to the output layer, and the connection weight can be obtained by ridge regression calculation. BLS has the following advantages: (1) BLS uses fewer layers than deep learning, and thus it has a simpler structure; (2) BLS uses ridge regression to calculate network weights, while DL often uses gradient descent. If the initial value setting is unreasonable, the DL needs more iterations and takes a longer time. (3) The input layer to MF, MF to EN, and MF and EN to the output layer weight of BLS are randomly assumed. The generated training parameters only have weight adjustment from the feature layer to the enhancement layer. Therefore, BLS requires fewer training parameters and labeled samples than DL. Zou et al. [13] propose a novel EEG multi-classification method by combining with BLS and a common spatial pattern. The result shows that its classification accuracy is better than the ELM and DBN algorithms, and the classification time is much faster. Recently, Shuang et al. propose a fuzzy broad learning algorithm [14]. The Takagi-Sugeno (TS) fuzzy system is embedded into the BLS to replace the MF in the original BLS with a set of TS fuzzy subsystems. The results indicate that fuzzy BLS outperforms other models including some state-of-the-art nonfuzzy and neuro-fuzzy approaches. On the basis of BLS, Jin [15] proposes another version of BLS based on graph regularization and uses it for face recognition, which has obvious performance improvements over BLS. The graph regularization uses the local invariance between data, in other words, similar images have similar performance in manifold learning. Han et al. [16] also propose a BLS algorithm based on manifold structure. The distinction with Jin is that the algorithm is mainly combined with the unified framework of non-uniform embedding. It is a dynamic system for predicting a large-scale chaotic time series. Liu et al. [17] apply broad learning and incremental learning into commonly used neural networks including radial basis function and multi-layer extreme learning machine and propose BLS-RBF and BLS-HELM algorithms.

However, BLS belongs to a supervised algorithm, and all of the above-mentioned algorithms are supervised methods. The quantity of unlabeled samples is far more than labeled samples in real life. The calibration process of labeled samples requires much labor, material, and financial resources. Therefore, semi-supervised learning is proposed to utilize unlabeled samples. In [18], a new safety-aware graph-based semi-supervised learning is proposed. The graph-based method is generally revealed by constructing a k nearest neighbors (k-NN) graph. Li et al. [19] propose a semi-supervised SVM for EEG signals classification. The algorithm could be used to reduce training effort and improve the adaptability of the P300-based BCI speller. Wulsin et al. [20] propose a semi-supervised deep confidence network algorithm for fast classification and anomaly measurement of EEG signals. The classification time of the method was found to be 1.7–103.7 times faster than the other high-performing classifiers. Jia et al. [21] propose a new semi-supervised deep

learning algorithm combined with the restricted Boltzmann machine and apply it for EEG signals classification. She et al. [22,23] improve the ELM algorithm and propose semi-supervised ELM and safe semi-supervised ELM for EEG signals classification respectively. The results show that classification accuracy has significant improvement over ELM. However, there are few applications of BLS to semi-supervised learning, thus BLS is extended to graph-based semi-supervised BLS (GSS-BLS) for classifying EEG signals in this paper.

The main contributions in this paper can be summarized as follows:

(1) Since the BLS algorithm belongs to supervised learning and can only use labeled samples, it is modified with semi-supervised learning to exploit both labeled and unlabeled data to find more useful information, contributing to improving the classification accuracy of original BLS with better generalization capability.
(2) The proposed GSS-BLS algorithm retains advantages of the original BLS algorithm, which can improve the classification accuracy compared with the traditional supervised learning.
(3) In the existing papers, BLS is often used for image classification and has not been involved in the field of biomedicine. We have improved it and broadened its application fields.

The remainder of this paper is organized as follows. Section 2 provides a description of the proposed semi-supervised BLS algorithm, including a brief introduction to the pre-processing of EEG and the principle of graph label propagation. Section 3 describes the performance of our method through a series of experiments on several motor imagery (MI) EEG datasets and Section 4 discusses GSS-BLS and the limitation of GSS-BLS. Finally, the conclusion and future works are presented in Section 5.

## 2. Materials and Methods

The classification process of EEG signals based on semi-supervised BLS is shown in Figure 1. It mainly includes three aspects:

(1) The original EEG signals are preprocessed, filtered by a Butterworth filter, and subjected to a dimensionality reduction process using the common spatial pattern (CSP) algorithm.
(2) The pseudo labels of the unlabeled EEG samples are obtained by the graph label propagation approach.
(3) The labeled samples and the unlabeled samples are sent to the BLS for training with corresponding labels and corresponding pseudo labels respectively, and then the GSS-BLS classifier is obtained and used to classify the testing set.

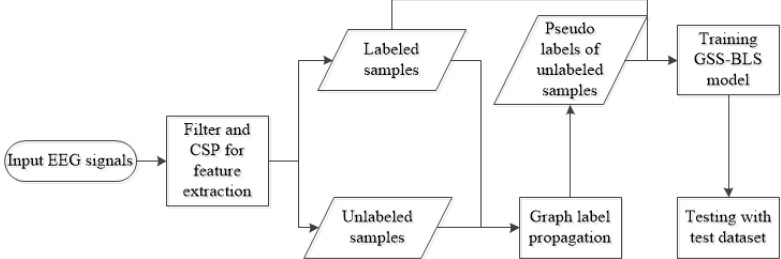

**Figure 1.** The flow chart of the proposed GSS-BLS (graph-based semi-supervised broad learning system) algorithm.

### 2.1. Common Spatial Pattern

CSP [24,25] is a common method for processing EEG signals. It belongs to the spatial domain filtering feature extraction algorithm and can extract spatial distribution components of each class from multi-channel brain-computer interface data. The basic principle of the CSP algorithm is using the

diagonalization of the matrix to find a set of optimal spatial filters for projection, and then the variance of the difference between the two classes of signals is maximized so that the feature vector with higher discrimination is obtained.

The CSP feature extraction of the 2-class EEG signals is briefly described below. Suppose $\mathbf{X}_1 \in N \times T$ and $\mathbf{X}_2 \in N \times T$ are multi-channel induced corresponding spatiotemporal signal matrixes of two motor imagery tasks. N is the number of EEG channels, T is the number of samples for each channel.

Find the covariance matrices after normalizing $\mathbf{X}_1$ and $\mathbf{X}_2$:

$$\mathbf{R}_1 = \frac{\mathbf{X}_1\mathbf{X}_1^T}{tr(\mathbf{X}_1, \mathbf{X}_1^T)}, \mathbf{R}_2 = \frac{\mathbf{X}_2\mathbf{X}_2^T}{tr(\mathbf{X}_2, \mathbf{X}_2^T)} \tag{1}$$

where $(.)^T$ represents transpose, $tr(.)$ represents the sum of matrix diagonal elements. Then find the mixed space covariance matrix and perform eigenvalue decomposition:

$$\mathbf{R} = \overline{\mathbf{R}_1} + \overline{\mathbf{R}_2} = \mathbf{U}\mathbf{\Sigma}\mathbf{U}^T \tag{2}$$

where $\overline{\mathbf{R}_i}(i = 1, 2)$ represents average covariance matrix, $\mathbf{U}$ is the eigenvector matrix of matrix $\mathbf{\Sigma}$. $\mathbf{\Sigma}$ is a diagonal array of corresponding eigenvalues. The eigenvalues are arranged in descending order to obtain a whitened value matrix, then $\mathbf{R}_1$ and $\mathbf{R}_2$ are transformed and analyzed by principal component decomposition:

$$\mathbf{S}_1 = \mathbf{P}\mathbf{R}_1\mathbf{P}^T = \mathbf{U}\mathbf{\Sigma}_1\mathbf{U}^T, \mathbf{S}_2 = \mathbf{P}\mathbf{R}_2\mathbf{P}^T = \mathbf{U}\mathbf{\Sigma}_2\mathbf{U}^T \tag{3}$$

where $\mathbf{P} = \sqrt{\mathbf{\Sigma}^{-1}}\mathbf{U}^T$. The transformation of the whitened EEG to the eigenvector corresponding to the largest eigenvalue in $\mathbf{\Sigma}_1$ and $\mathbf{\Sigma}_2$ is optimal for separating the variances in the two signal matrices. The spatial filter $\mathbf{W}_s$ corresponding to the projection matrix is:

$$\mathbf{W}_s = \mathbf{U}^T\mathbf{P} \tag{4}$$

With the matrix $\mathbf{W}_s$, the original EEG can be transformed into uncorrelated components.

$$\mathbf{Z} = \mathbf{W}_s\mathbf{X} \tag{5}$$

$\mathbf{Z}$ can be seen as EEG source components including common and specific components of different tasks.

### 2.2. Graph Label Propagation

The label information is smoothed over the graph via the graph label propagation algorithm [26,27]. The goal of the algorithm is to predict the labels of the unlabeled samples using both labeled data and unlabeled data. The algorithm can be described briefly as follows.

Suppose $\mathbf{F} = [\mathbf{F}_1; \mathbf{F}_2; \ldots; \mathbf{F}_{l+u}] \in \mathbb{R}^{(l+u) \times c}$ is a soft label matrix, $l$ and $u$ are numbers of labeled samples and unlabeled samples respectively together with the dimension of the row vector $c$, where each element in $\mathbf{F}_i$ belongs to $[0, 1]$ and $\mathbf{F}_i(i \in \{1, 2, \ldots, l + u\})$ is a row vector. Define matrix $\mathbf{Y} = [\mathbf{Y}_1; \mathbf{Y}_2; \ldots; \mathbf{Y}_{l+u}] \in \mathbb{R}^{(l+u) \times c}$ represents labels of samples and $\mathbf{Y}_i = [y_{i1}, y_{i2}, \ldots, y_{ic}]$. For labeled data $x_i$, if it is labeled as $j(j \in \{1, \ldots, c\})$, then $y_{ij} = 1$, otherwise $y_{ij} = 0$ [28].

Then the weighted graph $\mathbf{W}_g$ can be constructed. The relation of $x_i$ and $x_j$ is represented by $w_{ij}$:

$$w_{ij} = \begin{cases} e^{-\|x_i - x_j\|^2/\sigma_w^2} & if \begin{cases} x_i \in N(x_j) \\ x_j \in N(x_i) \end{cases} \\ 0 & otherwise \end{cases} \tag{6}$$

where $N(\bullet)$ is $k$ nearest neighbors of $x_i$ or $x_j$. $\|\bullet\|$ is Euclidean norm of a vector. $\sigma_w$ is a scaling parameter of Gaussian function [29]. Then construct the normalized Laplacian matrix $\mathbf{S} = \mathbf{D}^{-1/2}\mathbf{W}\mathbf{D}^{-1/2}$, in

which $\mathbf{D}$ is a diagonal matrix with its $(i, i)$-element equal to the sum of the $i$-th row of $\mathbf{W}_g$. Now the objective function is shown as below:

$$\min_{\mathbf{F}} \sum_{i,j} \|\mathbf{F}_i - \mathbf{F}_j\|^2 w_{ij} + \mu \sum_{i,j} \|x_i - x_j\|^2 w_{ij} \tag{7}$$

where the first term constrains the similar samples have similar labels and the second term constrains the graph optimization that similar features correspond to high similarities. $\mu$ is a trade-off parameter to balance weights between feature space and label space. In the optimization process, the existing method is fixing other variables and updating one at a time until it converges. So, while $\mathbf{F}$ is considered, the Equation (7) can be rewritten by:

$$\min_{\mathbf{F}} \sum_{i,j} \|\mathbf{F}_i - \mathbf{F}_j\|^2 w_{ij} = tr(\mathbf{F}\mathbf{S}\mathbf{F}^T), \ s.t. \ \mathbf{F}_l = \mathbf{Y}_l \tag{8}$$

Since $\mathbf{F}_l$ is the predicted labels of labeled points, we suppose $\mathbf{F}_l$ is real labels, in other words, $\mathbf{F}_l = \mathbf{Y}_l$. The differentiate of Equation (8) is shown below:

$$[\mathbf{F}_l \ \mathbf{F}_u] \begin{bmatrix} \mathbf{S}_{ll} & \mathbf{S}_{lu} \\ \mathbf{S}_{ul} & \mathbf{S}_{uu} \end{bmatrix} = 0 \tag{9}$$

where $\mathbf{F} = [\mathbf{F}_l, \mathbf{F}_u]$, $\mathbf{S}$ is rewritten as partitioned matrix. Now $\mathbf{Y}_l, \mathbf{S}_{lu}, \mathbf{S}_{uu}$ are known, and by calculating Equation (8) we can acquire $\mathbf{F}_u = -\mathbf{Y}_l\mathbf{S}_{lu}\mathbf{S}_{uu}^{-1}$.

### 2.3. Broad Learning System

The broad learning system mainly consists of three parts: mapping layer (feature nodes), enhancement layer (enhancement nodes), and output layer. The main algorithm is as follows.

Suppose a training set $\left\{(\mathbf{X}, \mathbf{Y}) | \mathbf{X} \in \mathbb{R}^{n \times d}, \mathbf{Y} \in \mathbb{R}^{n \times c}\right\}$, and $n$ is number of training samples, $d$ is characteristic dimension, $c$ is number of categories. Each training sample is represented as $x_i = (x_{i1}, x_{i2}, \ldots, x_{id})$ and the corresponding label is denoted as $y_i = (y_{i1}, y_{i2}, \ldots, y_{ic})$.

First, the training samples are mapped to the feature space $\mathbf{Z}^{N_w}$ through the feature mapping function $\phi_i, i = 1, 2, \ldots, N_w$. $N_w$ is number of mapped feature vectors.

$$\mathbf{Z}_i = \phi_i(\mathbf{X}\mathbf{W}_{e_i} + \beta_{e_i}), i = 1, 2, \ldots, N_w \tag{10}$$

where $\mathbf{W}_{e_i}$ is mapping weight matrix and $\beta_{e_i}$ is random deviation.

Second, define feature space $\mathbf{Z}^{N_w} \triangleq [\mathbf{Z}_1, \mathbf{Z}_2, \ldots, \mathbf{Z}_{N_w}]$ where $\mathbf{Z}_i$ is feature node, $i = 1, 2, \ldots, N_w$. Similar to feature nodes generated by training samples, feature nodes are also used for enhanced nodes.

$$\mathbf{H}_j \triangleq \xi_j(\mathbf{Z}^{N_w}\mathbf{W}_{h_j} + \beta_{h_j}), j = 1, 2, \ldots, m \tag{11}$$

where $\xi_j$ is a nonlinear activation function. The enhancement layer can then be represented as $\mathbf{H}^m \triangleq [\mathbf{H}_1, \mathbf{H}_2, \ldots, \mathbf{H}_m]$. In order to obtain a sparse representation of the training data and adjust the weight matrix $\mathbf{W}_{e_i}$ of the input layer to the output layer, BLS uses a linear function as the activation function of $\phi_i$ and $\xi_j$. So, BLS can be represented as:

$$\hat{\mathbf{Y}} = [\mathbf{Z}_1, \mathbf{Z}_2, \ldots, \mathbf{Z}_n, \mathbf{H}_1, \mathbf{H}_2, \ldots, \mathbf{H}_m]\mathbf{W}_{BLS} = \mathbf{A}\mathbf{W}_{BLS} \tag{12}$$

where $\mathbf{A} = [\mathbf{Z}^{N_w}, \mathbf{H}^m]$, $\mathbf{W}_{BLS}$ is a weight matrix of feature nodes and enhancement nodes to output layer. $\mathbf{W}_{BLS}$ can be optimized with the following objective function:

$$\underset{\mathbf{W}_{BLS}}{\operatorname{argmin}}(\|\mathbf{Y} - \mathbf{A}\mathbf{W}_{BLS}\|^2 + \lambda\|\mathbf{W}_{BLS}\|^2) \tag{13}$$

where the first term represents the training error and the second term is the regular term used to control the complexity of the model. $\lambda$ is a regular term coefficient used to balance the relationship between two terms. $\mathbf{W}_{BLS}$ can be obtained by simple derivation calculation.

$$\mathbf{W}_{BLS} = (\mathbf{A}^T\mathbf{A} + \lambda\mathbf{I})^{-1}\mathbf{A}^T\mathbf{Y} \tag{14}$$

### 2.4. Graph-Based Semi-Supervised BLS

The BLS method is subject to supervision and cannot use many unlabeled points. Therefore, combining the advantages of both BLS and graph label propagation, we propose GSS-BLS algorithm to achieve semi-supervised classification of EEG signals.

Assume a pre-processed training set $\left\{(\mathbf{X},\mathbf{Y})\middle|\mathbf{X} \in \mathbb{R}^{(l+u)\times d}, \mathbf{Y} \in \mathbb{R}^{(l+u)\times c}\right\}$ and corresponding labels are $\mathbf{Y}^l$ and $\mathbf{Y}^u$, where $\mathbf{Y} = [\mathbf{Y}^l, \mathbf{Y}^u]$. The weight matrix from the input samples to the feature vector is $\mathbf{W}_{Ms} = [\mathbf{W}_1, \ldots, \mathbf{W}_M]$ and random deviation is $\beta_{Ms} = [\beta_1, \ldots, \beta_M]$. Analogy Equation (10), the feature vector can be shown as:

$$\mathbf{Z}_{si} = \phi_i(\mathbf{X}\mathbf{W}_i + \beta_i), i \in \mathbf{M} \tag{15}$$

where M is number of feature vectors, $\phi(\bullet)$ is a nonlinear function, and different activation functions can be selected according to different situations. As with Equation (10), the linear function is still used here as an activation function.

After obtaining the feature space $\mathbf{Z}^S = [\mathbf{Z}_1, \mathbf{Z}_2, \ldots, \mathbf{Z}_M]$, the enhancement layer can be expressed as:

$$\mathbf{H}_{sj} = \phi_j(\mathbf{Z}\mathbf{W}_{sj} + \beta_{sj}), j \in \mathbf{N} \tag{16}$$

where $\mathbf{W}_{Ns} = [\mathbf{W}_1, \ldots, \mathbf{W}_N]$ is a random weight matrix and $\beta_{Ns} = [\beta_1, \ldots, \beta_N]$ is the deviation. Now the GSS-BLS can be represented as:

$$[\mathbf{Y}^l|\mathbf{Y}^u] = [\mathbf{Z}^S|\mathbf{H}_s]\mathbf{W}^m \tag{17}$$

where $\mathbf{W}^m$ is mapping layer and enhancement layer to output layer connection weight. The solution of $\mathbf{W}^m$ can be obtained by:

$$\underset{\mathbf{W}^m}{\text{argmin}}\|[\mathbf{Z}^s|\mathbf{H}_s]\mathbf{W}^m - [\mathbf{Y}^l|\mathbf{Y}^u]\|^2 + \lambda\|\mathbf{W}^m\|^2 \tag{18}$$

where $\lambda$ is a balance parameter and used to constraint $\mathbf{W}^m$. Equation (18) can be solved by ridge regression:

$$\mathbf{W}^m = (\lambda\mathbf{I} + [\mathbf{Z}^s|\mathbf{H}_s]^T[\mathbf{Z}^s|\mathbf{H}_s])^{-1}[\mathbf{Z}^s|\mathbf{H}_s]^T[\mathbf{Y}^l|\mathbf{Y}^u] \tag{19}$$

where $\lambda = 0$, Equation (19) degenerates into the least square problem, but if $\lambda \to \infty$, the solution is heavily constrained and tends to 0. So, we refer to BLS and set $\lambda = 2^{-30}$ [30]. By giving an approximation to the Moore-Penrose generalized inverse of $[\mathbf{Z}^s|\mathbf{H}_s]$, Equation (19) can be written as:

$$\mathbf{W}^m = [\mathbf{Z}^s|\mathbf{H}_s]^+[\mathbf{Y}^l|\mathbf{Y}^u] \tag{20}$$

Now the pseudo-inverse of $[\mathbf{Z}^s|\mathbf{H}_s]^+$ can be obtained:

$$[\mathbf{Z}^s|\mathbf{H}_s]^+ = \lim_{\lambda \to 0}(\lambda\mathbf{I} + [\mathbf{Z}^s|\mathbf{H}_s]^T[\mathbf{Z}^s|\mathbf{H}_s])^{-1}[\mathbf{Z}^s|\mathbf{H}_s]^T \tag{21}$$

Finally, the predictive labels can be written:

$$\mathbf{Y} = [\mathbf{Z}^s|\mathbf{H}_s]\mathbf{W}^m \tag{22}$$

Now the GSS-BLS algorithm for EEG classification can be summarized in Table 1.

**Table 1.** The specific steps of the GSS-BLS-based EEG (electroencephalogram) signal classification algorithm.

---

Algorithm 1: The GSS-BLS algorithm
Input: EEG signal preprocessed with CSP.

(a) Construct a Laplacian diagram according to Equation (6);
(b) Obtain pseudo-labels of the unlabeled samples according to Equation (9);
(c) Calculate feature nodes and enhancement nodes according to Equations (15) and (16);
(d) Calculate the connection weights $\mathbf{W}^m$ of the feature layer and the enhancement layer to the output layer according to Equation (20);
(e) Find the prediction labels using Equation (22) and the previously calculated parameters;

Output: Labels of predicted unlabeled samples.

---

## 3. Experiment and Analysis

### 3.1. BCI Datasets

In order to verify the validity and practicability of the GSS-BLS, we used three motor imagery EEG datasets of BCI competitions [31], including two 2-class datasets and one 4-class dataset, which were described as follows:

(1) Dataset IVa, BCI competition III [32]: The dataset contained EEG signals from five subjects, and each subject performed right hand and foot imaging tasks. EEG signals were recorded using 118 electrodes. The dataset of each subject included a training set and a testing set and the size of these datasets varied from person to person. More precisely, every subject performed 280 trials of experiments in which the subjects of A1, A2, A3, A4, and A5 were respectively composed of the training samples of 168, 224, 84, 56, and 28, with the remainder forming the testing set.

(2) Dataset IIIa, BCI competition III [33]: The dataset was formed of EEG signals from three subjects who performed left-hand, right-hand, foot, and tongue MI tasks. A 60-lead electrode was used to record the EEG signals. In order to highlight the two-category recognition performance, only two classes of EEG signals (left-hand and right-hand MI signals) were used as actual usage data. EEG signals contained 45 training and testing samples per class for subject B1 while subjects B2 and B3 had 30 training and testing samples per class respectively.

(3) Dataset IIa, BCI competition IV [34]: The dataset contained four classes of MI EEG signals from 9 healthy subjects (C1 to C9), who performed left hand, right hand, foot, and tongue imaging tasks. EEG signals were recorded using 22 electrodes in all experiments. The training set and the testing set contained 288 sets respectively.

In view of the particularity and complexity of the original EEG signals, it was necessary to perform EEG data preprocessing. For each subject, a time window of 0.5 $s$~2 $s$ was selected for EEG data extraction, and then a 5th-order Butterworth filter was used to perform band-pass filtering operation of 8~30 Hz [30]. Next, the EEG signals were reduced in dimension using the CSP algorithm. Finally, the processed EEG data was trained and tested by different algorithms.

### 3.2. Comparative Methods

To assess the performance of the proposed GSS-BLS on three EEG datasets, we investigated the following five methods for comparison.

(1) Supervised classifiers included ELM [35] and BLS [12]. The linear feature mapping was used in BLS and GSS-BLS and the linear kernel function was used in ELM in our experiments. The hyperparameter of ELM was selected through ten-fold cross-validation and the regularization coefficient of ELM was selected from $\{10^{-4}, 10^{-3}, \ldots, 10^3, 10^4\}$.

(2)   Semi-supervised classification methods were Squared-loss Mutual Information Regularization (SMIR) [36] and Laplacian SVM (LapSVM) [37]. SMIR applied the Gaussian kernel and the kernel width was the median of all pairwise distances times the best value among {1/15,1/10,1/5,1/2,1}. The linear kernel function was also used for LapSVM.

(3)   The special classifier was HELM [38] for it was supervised with a multilayer structure.

### 3.3. Experimental Results

In order to evaluate the performance of GSS-BLS, two performance indexes were considered: the kappa value [39] on each subject and average kappa value for the classification of testing samples. The higher the kappa value was, the better the classification result we would get. For supervised methods, only the labeled samples were used to train the classifier and the trained classifier was used to predict the labels of unlabeled samples. For semi-supervised methods, all labeled and unlabeled samples were used to train the classifier. Since we experimented with the different proportions of training samples, the ratio of labeled to unlabeled samples was 1:4 which could achieve a satisfying performance. The comparing methods were also trained to achieve the best results. The performance was evaluated in terms of the mean kappa value and standard deviation (kappa ± std) using $10 \times 10$-fold cross-validations. The performance of the Dataset IVa was shown in Table 2, IIIa in Table 3, and IIa in Table 4.

**Table 2.** Kappa value on testing data of BCI (brain-computer interface) Competition III Dataset IVa.

| Dataset (All/Test) | BLS kappa ± std | ELM kappa ± std | HELM kappa ± std | SMIR kappa ± std | LapSVM kappa ± std | GSS-BLS kappa ± std |
|---|---|---|---|---|---|---|
| A1(280/112) | 0.201 ± 0.035 | 0.343 ± 0.026 | 0.244 ± 0.018 | 0.201 ± 0.001 | **0.445 ± 0.073** | 0.323 ± 0.064 |
| A2(280/56) | 0.964 ± 0.001 | **1** | 0.944 ± 0.001 | **1** | 0.961 ± 0.011 | 0.968 ± 0.011 |
| A3(280/196) | 0.193 ± 0.014 | 0.163 ± 0.024 | 0.243 ± 0.023 | 0.200 ± 0.001 | **0.321 ± 0.152** | 0.227 ± 0.064 |
| A4(280/224) | 0.477 ± 0.007 | 0.558 ± 0.020 | 0.470 ± 0.018 | 0.413 ± 0.001 | 0.441 ± 0.171 | **0.689 ± 0.047** |
| A5(280/252) | 0.704 ± 0.001 | 0.660 ± 0.008 | 0.706 ± 0.008 | 0.530 ± 0.001 | 0.675 ± 0.245 | **0.738 ± 0.029** |
| Average | 0.508 ± 0.012 | 0.545 ± 0.016 | 0.521 ± 0.014 | 0.469 ± 0.001 | 0.569 ± 0.130 | **0.589 ± 0.043** |

**Table 3.** Kappa value on testing data of BCI Competition III Dataset IIIa.

| Dataset (All/Test) | BLS kappa ± std | ELM kappa ± std | HELM kappa ± std | SMIR kappa ± std | LapSVM kappa ± std | GSS-BLS kappa ± std |
|---|---|---|---|---|---|---|
| B1(90/45) | 0.887 ± 0.007 | 0.907 ± 0.001 | 0.889 ± 0.001 | 0.844 ± 0.001 | 0.928 ± 0.030 | **1** |
| B2(60/30) | 0.143 ± 0.016 | 0.160 ± 0.016 | 0.143 ± 0.016 | 0.133 ± 0.001 | 0.163 ± 0.124 | **0.170 ± 0.123** |
| B3(60/30) | 0.943 ± 0.016 | 0.933 ± 0.011 | 0.977 ± 0.016 | **1** | 0.909 ± 0.079 | **1** |
| Average | 0.658 ± 0.117 | 0.667 ± 0.009 | 0.670 ± 0.011 | 0.659 ± 0.001 | 0.667 ± 0.078 | **0.723 ± 0.041** |

**Table 4.** Kappa value on testing data of BCI Competition IV Dataset IIa.

| Dataset (All/Test) | BLS kappa ± std | ELM kappa ± std | HELM kappa ± std | SMIR kappa ± std | LapSVM kappa ± std | GSS-BLS kappa ± std |
|---|---|---|---|---|---|---|
| C1(576/288) | 0.566 ± 0.012 | 0.587 ± 0.022 | 0.589 ± 0.011 | 0.537 ± 0.001 | 0.610 ± 0.124 | **0.615 ± 0.013** |
| C2(576/288) | 0.253 ± 0.011 | 0.297 ± 0.028 | 0.276 ± 0.016 | 0.227 ± 0.001 | 0.334 ± 0.094 | **0.337 ± 0.033** |
| C3(576/288) | 0.671 ± 0.020 | 0.648 ± 0.017 | 0.706 ± 0.008 | 0.699 ± 0.001 | **0.777 ± 0.061** | 0.690 ± 0.010 |
| C4(576/288) | 0.337 ± 0.015 | 0.360 ± 0.018 | 0.355 ± 0.007 | **0.394 ± 0.001** | 0.363 ± 0.092 | 0.387 ± 0.037 |
| C5(576/288) | 0.146 ± 0.007 | 0.174 ± 0.020 | 0.178 ± 0.008 | 0.167 ± 0.001 | **0.280 ± 0.072** | 0.182 ± 0.011 |
| C6(576/288) | 0.282 ± 0.012 | 0.272 ± 0.015 | 0.267 ± 0.013 | 0.264 ± 0.001 | **0.286 ± 0.100** | 0.275 ± 0.036 |
| C7(576/288) | 0.706 ± 0.014 | 0.700 ± 0.018 | 0.726 ± 0.010 | 0.708 ± 0.001 | 0.695 ± 0.075 | **0.728 ± 0.008** |
| C8(576/288) | 0.652 ± 0.017 | 0.697 ± 0.017 | 0.700 ± 0.018 | 0.685 ± 0.001 | 0.748 ± 0.069 | **0.773 ± 0.006** |
| C9(576/288) | 0.560 ± 0.017 | 0.573 ± 0.025 | 0.605 ± 0.020 | 0.644 ± 0.001 | 0.719 ± 0.089 | **0.739 ± 0.014** |
| Average | 0.464 ± 0.014 | 0.479 ± 0.019 | 0.488 ± 0.013 | 0.481 ± 0.001 | **0.534 ± 0.086** | 0.525 ± 0.018 |

Table 2 showed that GSS-BLS yielded the highest kappa value in two subjects (A4, A5) and average mean kappa value (0.589). In A4 and A5 subjects, GSS-BLS improved significantly compared with other algorithms. The LapSVM approach achieved the higher kappa values in subjects A1 (0.445) and A3 (0.321) but both of which were very low among the five algorithms from Table 2. The main reason for this, we predicted, was that the datasets of A1 and A3 were terrible. Comparing A3 with A5 showed that the size of testing was similar while the results of the two subjects differed widely. So, we suspected there was something wrong with the datasets of A1 and A3 rather than five methods. Compared with the other four subjects, the kappa value of subject A2 was high in five methods due to more training samples and fewer testing samples, especially since the kappa values of ELM and SMIR reached 1, which meant that the classification was completely correct. The average mean kappa values showed that SMIR was the worst in the six methods. This might be due to the fact that SMIR was a multi-class probability classification based on square loss mutual information regularization, which was mainly to maximize the class probability output to classify unlabeled samples. Along with the various interference information, the gap of EEG signal data was not so large that it would have a certain impact on the probability calculation.

Table 3 showed that GSS-BLS achieved the highest kappa value in subjects B1, B2, and B3, as well as the average mean kappa value (0.723). In subject B1, the GSS-BLS results were significantly better than the other five algorithms, especially the three supervised algorithms. Although GSS-BLS achieved the best performance, the kappa values of subject B2 were very low in all algorithms which were similar to A3. The mean kappa values of SMIR and GSS-BLS achieved 1 in subject B3. It could be seen that GSS-BLS yielded a slightly higher average mean kappa value compared to other methods and the comparison methods did not differ much in average mean kappa value.

From Tables 2 and 3, we concluded that the proposed GSS-BLS algorithm had good classification results in the 2-class EEG datasets. In order to further verify the performance of GSS-BLS, it was tested in the 4-class EEG dataset and compared with other algorithms. The results were shown in Table 4.

Table 4. showed that the GSS-BLS algorithm achieved the best results in five subjects (C1, C2, C7, C8, and C9) and the LapSVM approach performed the best in three subjects (C3, C5, C6) and SMIR reached the best in subject C4. For subject C5, the kappa value of the LapSVM algorithm was significantly better than other algorithms. The reason for this could be that perhaps the graph constructed for LapSVM was better in subject C5 and used more unlabeled information. In subjects C1 and C2, the kappa values of GSS-BLS were small gaps compared to other algorithms while significantly better than other algorithms in subjects C7, C8, and C9. From Table 4, the kappa values of three semi-supervised algorithms were slightly higher than three supervised algorithms except that the SMIR was lower than HELM in average mean kappa value. In terms of the average mean kappa value, LapSVM performed the best result whereas the standard deviation was higher than other methods. GSS-BLS was only 0.007 lower than LapSVM and stable for classifying EEG signals from standard deviation. Moreover, among the nine subjects, GSS-BLS achieved better results than BLS except for subject C6.

In summary, GSS-BLS achieved better classification results in the 4-class EEG dataset. The experimental results above show that the proposed GSS-BLS gave always better results than comparison methods. This could be explained that the GSS-BLS model used unlabeled data and provided useful additional information. Consequently, thanks to this comparison, a positive behavior of the graph label propagation was observed.

### 3.4. Algorithm Performance with Different Proportions of Training Samples

In addition to the comparisons of various algorithms on different datasets, we also considered that the semi-supervised algorithm would be affected by the number of training samples. Therefore, we conducted an evaluation of the proportion of training samples for each subject and the experimental results showed that our proposed GSS-BLS and other semi-supervised algorithms outperformed the supervised algorithms in the case of a small proportion of training samples. Since all subjects presented

similar regularity in the experiments of different proportions of training samples, the results for four representative subjects (B1, A2, C8, C9) were shown in Figure 2.

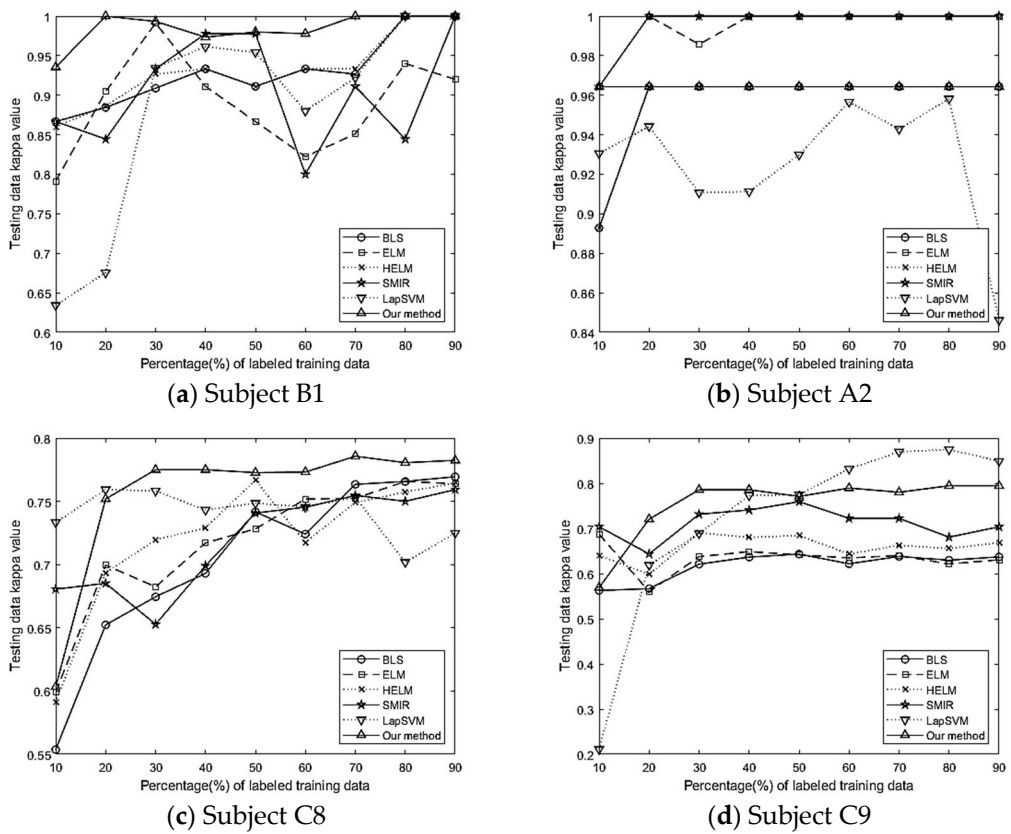

**Figure 2.** Kappa values of the algorithms for four subjects with different proportions of labeled samples.

As shown in Figure 2, under different conditions of training samples on the 2-class and 4-class datasets, the kappa values of the testing sets showed that semi-supervised methods were better than supervised algorithms. In subjects B1 and A2, GSS-BLS outperformed comparison approaches at 10% to 90% of the training samples. In addition, when the number of training samples was less than 30%, the kappa value of GSS-BLS was obviously higher than other algorithms in Figure 2a. When the proportion of training samples was above 80%, the kappa values of various algorithms increased significantly. From subject A2, we could find that algorithms were relatively stable except for the obvious fluctuation of LapSVM. The kappa value of the GSS-BLS algorithm was maintained at around 0.965. Although it was not comparable to ELM and SMIR, the disparity was not particularly obvious. Figure 2c showed GSS-BLS achieved the best under different ratios of training samples except that it was lower than LapSVM and SMIR algorithms at the ratio of 10%. The GSS-BLS was superior to other algorithms when the ratio was less than 50% while LapSVM was better when it was above 50% in Figure 2d.

Generally, the results showed that the GSS-BLS outperformed the other algorithms in small training samples since the GSS-BLS exploits the underlying manifold structure of the labeled and unlabeled data space. However, the performance of GSS-BLS, as well as other methods, sometimes degraded with the ratio of labeled samples. To our best understanding, the reasons might be that the impact of labeled samples would increase as the labeled ones increased, and the labeled ones might degrade the effectiveness of the GSS-BLS since the information of some inappropriately labeled ones would mislead the process of training.

## 3.5. Parameter Analysis

There were three main influenced parameters in this paper, the parameters in Equation (7) and the number of feature nodes and enhancement nodes of GSS-BLS. In this paper, the parameters of each subject were analyzed. Since the parameter analysis of 2-class was simpler than 4-class, we only presented the results of four classifications. Corresponding to Section 3.4, we only gave the analysis of the parameters of subjects C8 and C9.

As shown in Figure 3, the area of kappa value was stable when $\mu \geq 60$ in subject C8 and the fluctuation was little. The range of best kappa value affected by $\mu$ was greater than 45 and less than 55. Compared to subject C8, the value of $\mu$ had a great influence on the kappa value in subject C9, but when $\mu \geq 60$, the kappa value fluctuated drastically but did not reach the highest. $50 \leq \mu \leq 60$ was the considerable range to perform better kappa value. It could be found from the two subjects that the size of the value had a certain impact on performance. Considering all results of subjects, we chose $50 \leq \mu \leq 55$ for the experiments.

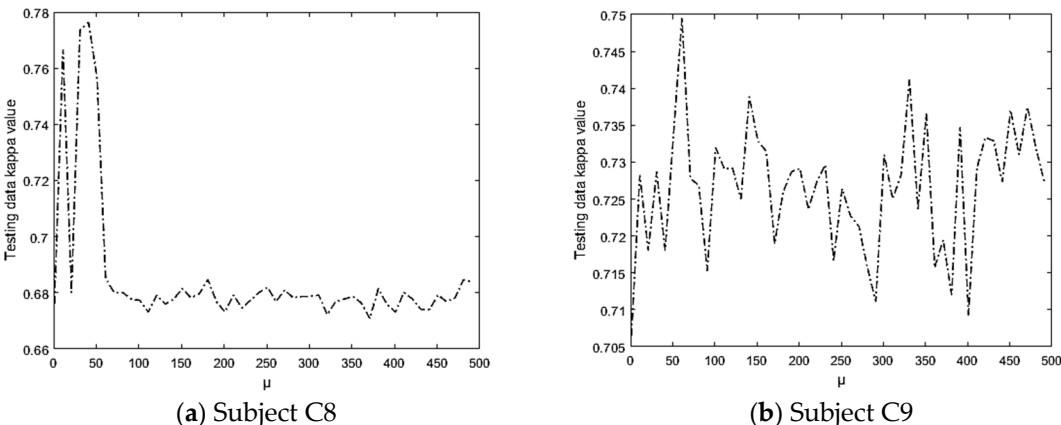

(**a**) Subject C8　　　　　　　　　　(**b**) Subject C9

**Figure 3.** Value test of GSS-BLS with different values for 2 subjects.

As shown in Figure 4, as the feature nodes and enhancement nodes increased so did the kappa value in subject C8. When the feature nodes and the enhancement nodes were in the range of 90–100, the kappa value achieved best and tended to be smooth. In subject C9, the result of kappa value showed a decreasing trend with the increasing feature nodes, but the kappa value had little effect as the enhancement nodes increased. The optimal results were achieved when feature nodes were in 10–20 and enhancement nodes in 90–100. In general, when the enhancement nodes increased, the kappa value increased, and the classification effect was improved.

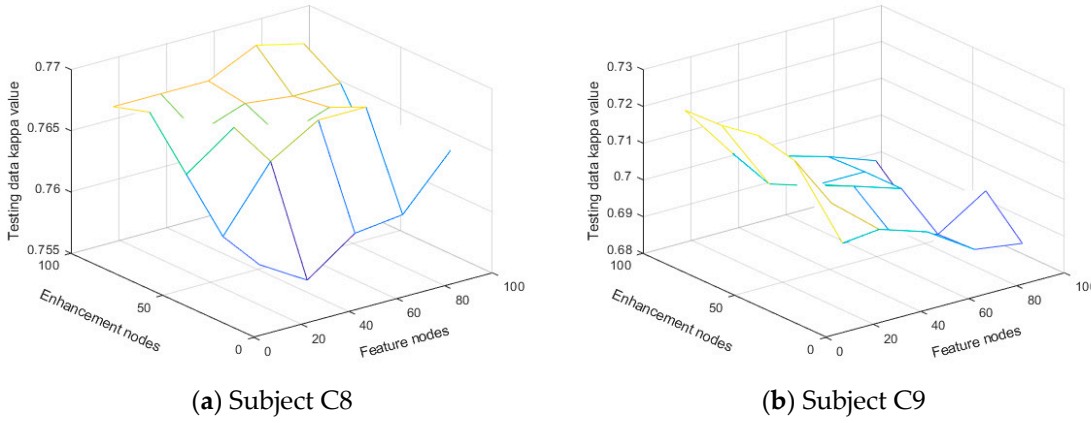

(**a**) Subject C8　　　　　　　　　　(**b**) Subject C9

**Figure 4.** Kappa value test of GSS-BLS with different feature nodes and enhanced nodes in two subjects.

## 4. Discussion

In the experiments, the proposed method has achieved better results compared to the other five methods in the classification of EEG signals. When compared with supervised algorithms, GSS-BLS has better performance than supervised methods, such as ELM, BLS, and HELM, which confirms that GSS-BLS uses unlabeled samples with additional information. However, compared with the other two semi-supervised algorithms, GSS-BLS (0.525) is lower than LapSVM (0.534) but greater than SMIR (0.481) of the average kappa value in the 4-class EEG dataset, but LapSVM algorithm is significantly higher than GSS-BLS in standard deviation, which is less stable than the GSS-BLS. GSS-BLS achieves the best results in five subjects whereas LapSVM yields optimal results in three subjects. The main reasons why GSS-BLS loses to LapSVM in three subjects can be boiled down to the following. The graph constructed for LapSVM was better in three subjects, and the original EEG data is preprocessed simply, so we might reserve the main information of motor imagery including artifacts. Therefore, the irrelevant factors are more influential in GSS-BLS than LapSVM. Overall, GSS-BLS is just below LapSVM in average mean kappa value. Therefore, GSS-BLS also achieves good classification results in the classification of the 4-class EEG dataset. From the experiment of different proportions of training samples, we can find that GSS-BLS achieves satisfying performance with limited labeled samples, so it solves the situation that the scale of EEG signals is small and the cost for labeling EEG signals is massive.

There are some limitations in the classification of EEG signals. GSS-BLS is trained offline, so it may result in false classification when used in online applications. GSS-BLS is applied for EEG signals, the scale of which is small, so the feasibility of this usage for big data is doubted. In addition, it defaults when adding training samples, and the classifier will be more stable with better classifying results, but actually, the insecurity of an increased sample is not considered.

## 5. Conclusions

In this paper, we propose a novel graph-based semi-supervised algorithm for the classification of EEG signals. The assessing of GSS-BLS is performed in 2-class as well as 4-class motor imagery with five other comparison algorithms. In addition, we also analyze the relevant parameters of GSS-BLS, and the kappa value affected by the different ratios of training samples. The results show that GSS-BLS yields better performance in three datasets, especially in the 2-class set. The average mean kappa value of GSS-BLS is better than other algorithms, except slightly lower than LapSVM in the 4-class dataset. Compared with BLS and other supervised methods, GSS-BLS offers significant improvements since it uses unlabeled samples and extracts additional information. In the case of reducing the number of labeled samples, GSS-BLS can also obtain better classifying results, which can reduce the cost of labeling samples. However, GSS-BLS has some disadvantages which we do not consider in the experiments. When the classifier for GSS-BLS is trained, we assume the increase of unlabeled samples will optimize the performance of the classifier, but actually the addition of unlabeled samples may lead to a decrease in performance [18]. Furthermore, the similarity of data structure for the training model is not considered, which is one of the factors that affects the performance of the classifier [40]. In future works, we will consider the security of unlabeled samples and the potential internal structure among data, and it will serve to improve the classification and evaluation of motor imagery EEG signals.

**Author Contributions:** Q.S. and Y.Z. conceived and designed the research; Y.Z. performed the research and made analysis; Q.S. and Y.Z. wrote the draft; H.G., Y.M., and Z.L. offered discussions and revisions.

**Acknowledgments:** This work is supported by National Natural Science Foundation of China (Nos. 61871427 and 61671197).

**Conflicts of Interest:** The authors declare no conflict of interest.

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
