# Peer review of "Decoding EEG in Motor Imagery Tasks with Graph Semi-Supervised Broad Learning"

_electronics, doi:10.3390/electronics8111273_

Round 1
Reviewer 1 Report
The authors are dealing with an up-to-date problem, to integrate lots of unlabelled data to comparably few manually labeled data sets. The inflated data base allows both the use extensive supervised AI techniques and may lead to an increase in the overall performance in classification tasks. The topic catches my interests and can be rated as very important in the century of unlabelled flood of data.
To deal with a mixture of labeled and unlabelled data the authors present a method called 'Graph-based Semi-Supervised Broad Learning System' and demonstrated it's use and performance on publicly available EEG data for specific brain computer interface tasks.
Although the presented method could outperform established methods, my overall impression of the paper is a bit disappointing. I noticed a lot of weaknesses described below and encourage the authors to continue and improve their work in this field:
Overall structure
The paper is not structured according to the 'Authors instructions', as a clear separation of experimental, statistical methods and the final results is missing. Also the technical description is hard to follow and the authors original work is hard to identify. According to the 'Authors instructions': "well-established methods can be briefly described and appropriately cited" - so please focus on the GSS modification of the BLS. Also the name and version of any software used and available computer code is not included in the manuscript.
Abstract
Please be concise.
L12 ''its performance can be determined by identifying of electroencephalogram (EEG) accurately and rapidly" -- What do you mean?
L14 "unlabeled samples are more than labeled samples" -- this and the following two sentences are telling the same.
Please write "Cohen's kappa"
Introduction
L38: "It can turn" - What do you mean?
L39: "to control external devices by pre-processing, feature extraction and classification" - consider revising this sentence
L41: "It is still a difficult point to identify EEG signals correctly and quickly." - I think you mean the correct manual labelling from the EEG can be difficult. How is this done? Searching for action potentials?
L55: et al, Deep Belief Net
Concrete performance values in the supervised and semi-supervised methods are not provided.
L98: You are not identifying EEG signals - you are classifying/labelling EEGs. Materials and Methods
The notation of the methods are not nicely presented, are lacking of information and consequent use of notation is often missing.
L117: CSP abbreviation not listed before.
Figure 1: try to preserve 'unlabeled' as a one-liner
L132: Why not defining it for a multi-class problem?
L134: What do you mean with "two types of motion imaging tasks"
L138: please write: (.)T, tr(.)
L154: The same subsection header as before
L158: what is l, u and c?
L159: I guess c is the dimension of the row vector?
L159: 'i' is discrete, so please don't use the interval notation, use set notation instead
L160: Y should be a matrix with Y1,1 to Yl+u,c ?
L160: xi is not defined
L163: Try to avoid repetitions and simplify the sentences, e.g. 'The relation of xi and xj is represented by wij: [Formula]'
L165: What kind of neighbors? How do you define the neighborhood? Is this KNN on Euclidean distances?
L165: Please write Euclidean norm instead of "2-norm". Please remove the subscript 2 in the following formulas, since you already wrote that it is the '2-norm'.
L165: sigma is the standard deviation - if there is a specific name / scientific term use that consequently.
L174: I don't see the optimization. It's followed by the results in line 175.
L175: What happened to mu? Set to zero?
L176: trace is already defined
L184: New dimensions N,D and C? Isn't C the same than c?
L188: When talking of the feature space, use the notation ZNw directly there.
L200: Could it be shorter: \hat{Y} = [ZNw,Hm]WBLS ?
L213: S and U stands for supervised and unsupervised sets? Is this new or already defined in literature?
L215: Ws defined here and in L224
L215: Where is the deviation coming from?
L221: Zs is different to L192?
L232: If the solution tends to zero if lambda goes to infinity, why then set to 2-30? Also lambda=1 is far away from infinity. Can't you handle lambda as a hyperparameter which will be optimized in the training process?
L243: Please add the BCI dataset description and evaluation metrics here and name the methods and implementation of comparable methods.
Experiment and Analysis
L245: A paper cannot select data sets
L246: For method validation?
L253: Are all the sets (training and test sets) labelled? I don't understand why there are different sizes of training data. I think it's better to use an own split.
What are the record lengths? What was the sampling frequency.?
L265: Please use SI-units: s for seconds. How was the time window defined? How do you handle different data lengths?
L271: BLS and ELM stands for? References are missing.
L272: "The supervised algorithms", Which? Combine with previous sentence.
L274: Do you removed the true labels? How do you actually conducted that to get a 1:4 ratio? Randomly and repeatedly?
L274: "Other algorithms adjusted the parameters to the best condition" What is meaning? They are trained to optimizè Kappa?
L279: What is the evaluation data? A hidden test set or a validation set. Please clarify the data structure and actual work flow to allow reproducibility.
Where are the CV results? Add the size of data in the first column.
L288: "However, the kappa of the other four algorithms were not low. The reason was that 24 test points for subject A2 and the train points is 224." Why have you not tried to use some of the training sets for testing?
L292: Is this difference statistically significant?
L342: spacing
342: Please put Figures 2 and 3 together. Figure: When the percentage of labeled training data increases the testing kappa should in my opinion not decrease if 10 time repeated 10-fold- cross-validation was carried out correctly.
How do you carried out this study? Removed labels randomly?
How does it look for other probands? Better try to summarise all the patients in one figure.
L383: Please include the CV-SE in the Figure.
L391: Which parameter have you used for mu? 50 or 55 oder in-between?
L390: Have you fixed the seed to check the effect of mu to fix the CV -folds?
L405: "s, such as ELM, BLS and HELM,, such as ELM," ???
Discussion/Conclusions
Limitations of the method are not discussed: unbalanced data, data size.
How is the performance on large data sets? How is the performance on other classification tasks? Do you think it's usable in other tasks? How large is the field of application?
Reviewer 2 Report
The authors present an interesting paper on using a novel classification algorithm to address the issues associated with having a data set that is lacking labels and finding ways to use unsupervised learning. The paper takes three different EEG datasets and contrasts the performance of their algorithm with several other known metrics.
The authors should address a few key issues prior to the paper being ready for publication:
Firstly, the paper overall should be proofread and checked for grammar errors.
Secondly, the paper lacks motivation and background. Specifically:
- Why use EEG signals?
- How are EEG signals obtained?
- What is the state of the art?
- How is this an improvement?
Is the filtering and dimensionality reduction process applied to all EEG signals? This means that the algorithm sees both the training and test sets which leads to data contamination between the two data sets.
Why does each experiment have such different numbers of electrodes? Does this play a role?
The tasks performed by each subject should be better explained. What is a "right hand and foot imaging tasks"?
Kappa value statistic should be defined or referenced.
Concerning that the machine learning algorithms completely fail for certain subjects. A1, A3, B2, C2, C4, C5, C6. Can the authors explain why or perhaps offer some explanation or thoughts on the robustness of these algorithms to different subjects?
Figures 2 and 3 should be improved for readability. Clearer legend, axis labels, and colour to plots.
Authors should explain why these chose the specific subjects to discuss in Figures 2 and 3.
Reviewer 3 Report
In this paper, a new semi-supervised algorithm for classifying EEG signals is proposed. This algorithm uses CSP, Graph Label Propagation and BLS. BCI is its main application.
Comments and questions:
1) In line 165, you say “ is a parameter of Gaussian function”. I would say is rather a scaling parameter. What is your choice for this parameter? For example, Ref. https://doi.org/10.1016/j.eswa.2016.08.059 uses a local scaling defined as a distance σi=d ( xi , x N) where xN is the N’th neighbour of data point xi. (And N=7).
2) In line 265: “ it is necessary to perform EEG data preprocessing”. What about the artifacts? Did you filter them?
3) When you analyze your results, you do not include an analysis between BLS and GSS-BLS. I think It could be interesting to say that GSS-BLS gives always better results than GLS. This could be explained because the GSS model uses more data and provides useful additional information. Consequently, thanks to this comparison, a positive behaviour of the graphic label propagation is observed. What do you think?
4) Using GSS-BLS, you get similar or better results than other methods. The results of the classification are acceptable and seem logical. But, are your results good enough to say that GSS-BLS is a good method to be implemented in a BCI? can you use your algorithm in an online application? Any problem?
In addition, GSS uses a defined set of labels. But in a BCI, an unlabeled sample that does not belong to the GSS label set might appear. What does your method do? Does your method force that sample to be labeled with a label from the GSS set?
Minor comments:
5) In line 18 you say: “graph-based semi-supervised broad learning system (GSS-BLS),“. I think it is not necessary to repeat it in line 121 and 212.
6) In line 110, the concept of "graphic label propagation" appears for the first time. It would be better if you introduce this concept before (introduction + Ref.).
7) In line 126 you introduce the concept: “Common spatial pattern (CSP) [23-24]”. You need to do it before because CSP is used in line 73 y 117.
8) In line 154: “2.2. Common spatial pattern” → You need to change it! It is the same than 2.1.
9) Do not use the same letters to explain different algorithms if those letters represent different things.
10) In line 276: “the mean kappa”. It would be better to add a ref.
11) Review your text. For example, I do not understand the paragraph in line 414: "The ratio of the number of different training sample experiments, the proposed method in a small sample of the training would be able to achieve the accuracy of the time with all the training samples involved classifier training, which made the training can be classified under the EEG have fewer tag number of samples of the situation. "
Round 2
Reviewer 2 Report
The reviewer would like to thank the authors for addressing the majority of the comments.
However, I believe that some discussion is necessary on the high variation in performance across subjects. Of course, tuning parameters, raw-data, feature extraction process will all affect the prediction outputs. But it is important to see why it fails completely for select patients. This implies that this method is not robust, training is not indicative of future performance, and we cannot use such a method in real world applications where it fails completely for 7/17 patients.
Some discussion or comparison of the features and/or the raw data for successful prediction and failures should be included.
Additionally, the reviewers have not fixed the majority of the grammatical errors from the previous iteration. These must be addressed as there are quite a few throughout the document.
Reviewer 3 Report
A) In my previous review, I had advised the authors to review the entire article. I think the paper has hardly improved. I suggest the authors review their work with the help of a native English speaker.
B) I asked you about the parameter σ (L172). Please provide a ref in the paper that justifies your choice.
C) I asked you about the presence of artifacts in the EEG signals. Normally, eye movements, neck movements, etc. produce artifacts in the EEG signals. You used a band-pass filter that passes frequencies between 8 and 30 Hz. Did you use this filtering to remove potential artifacs? If so, it is hard to believe that with this filtering you have corrected all possible artifacts. However, using this filtering, important information related to brain activity is also lost. Please justify your choice or provide a reference where such filtering has been successful for BCI, either to eliminate artifacts or to select meaningful information (or both simultaneously).
D) A good example where you should improve your text is your answer “Since there is no perfect thing, GSS-BLS also has some limitation in classification of EEG signals. First, GSS-BLS is trained in offline so the method may not classify correct results if it is in an online application. Second, GSS-BLS uses for EEG signals, and we know the size of EEG signals always small, so we doubt whether the method can be applied for big data. Third, we default when training samples is added, the classifier will be more stable and have better classifying results, but in fact we do not estimate the safety of an increased sample.”
Please, avoid expressions like "Since there is no perfect thing" etc. I suggest again the authors review their paper with the help of a native English speaker.
The training process is offline, but is there a problem to use your trained model in an online application? For example, GSS uses a defined set of labels. But in a BCI, an unlabeled sample that does not belong to the GSS label set might appear. I am referring to a sample that did not participate in the training process. What does your method do? Does your method force that sample to be labeled with a label from the GSS set?
Round 3
Reviewer 3 Report
OK about the latest answers.
I think this document is much better than the first version. However, I insist on the sentence: "It is well known that nothing is perfect in the world, so is GSS BLS." I think it adds no value to the paper. I would remove it.
Author Response
We thank the reviewer for the valuable suggestion. We agree the suggestion of reviewer and delete the sentence: “It is well known that nothing is perfect in the world, so is GSS-BLS.”